

# Predicting soil moisture across a heterogeneous boreal catchment using terrain indices

Johannes Larson[1], William Lidberg[1], Anneli M. Ågren[1], Hjalmar Laudon[1]

[1]Department of Forest Ecology and Management, Swedish University of Agricultural Sciences, Umeå, 901 83, Sweden

*Correspondence to*: Johannes Larson (Johannes.larson@slu.se)

**Abstract.** Soil moisture has important implications for drought and flooding forecasting, forest fire prediction and water supply management. However, mapping soil moisture has remained a scientific challenge due to forest canopy cover and small-scale variations in soil moisture conditions. Digital terrain analysis has been suggested as a way forward to model soil moisture variability across landscapes, and multiple digital terrain indices have been developed. However, the performance of these terrain indices depends on the resolution of the digital elevation models used and, in many cases, user-defined index-specific thresholds. In this study, we compared soil moisture predictions using nine different terrain indices and available soil wetness maps, at varying resolutions and user-defined thresholds, with a field dataset of soil moisture registered in five classes from a forest survey covering a boreal landscape. We found that topography could explain the spatial variation in soil moisture conditions but the effects of changing DEM resolution and user-defined thresholds severely affected the performance of the soil moisture modelling. These results demonstrate that modelled soil moisture conditions need to be validated, as selecting unsuitable DEM resolution or user-defined threshold can give ambiguous and even incorrect results. Challenges caused by heterogeneous soil types within the study area highlight the need for local knowledge when interpreting the modelled results.

## 1 Introduction

Soil moisture represents plant available water at the land surface that is not derived from groundwater, rivers and lakes, but instead in the pores of the soil. It consists of unsaturated soil, affected by variable temporal and spatial dynamics that regulate fundamental ecosystem functions such as plant growth, nutrient cycling and carbon accumulation (Olsson et al., 2009; Högberg et al., 2017; Wang et al., 2019). Soil moisture also has important implications for drought and flooding forecasting, forest fire prediction and water supply management (Koster et al., 2010; Robock, 2015; O et al., 2020). While temporal variability in soil moisture is largely determined by precipitation, temperature and soil characteristics, topography acts as a first order control of spatial variation in soil moisture within most landscapes (Florinsky, 2016).

Predicting soil moisture patterns across space and time remains an important scientific challenge, limited by large temporal variability, small-scale heterogeneous responses to precipitation inputs and local soil properties. While small-scale



spatial variability often limits the use of empirical measurements for upscaling, temporal dynamics superimposed on such heterogeneous patterns create an additional challenge. Due to the effect that topography has on the spatial variation in soil moisture conditions, such information is a fundamental part of soil moisture modelling. A Digital Elevation Model (DEM) is a digital representation of a terrain surface, often generated using remote sensing techniques such as photogrammetry or airborne Light Detection And Ranging (LiDAR). Terrain indices extracted from DEMs have become widely used in soil and

hydrologic sciences predicting surface water and groundwater flow paths, and soil moisture conditions.

An early and successful approach to modelling soil moisture conditions was the Topographic Wetness Index (TWI) developed by (Beven and Kirkby, 1979). TWI is a function of both the slope and upslope contributing area, and is still widely used in landscape modelling. TWI has been shown to be sensitive to DEM resolution (Western et al., 1999; Sørensen and Seibert, 2007; Lin et al., 2010) and the specific flow algorithms used (Sørensen et al., 2006; Kopecký et al., 2021). TWI has

been followed by several other terrain indices based on similar approaches such as the Downslope index (DI) (Hjerdt et al., 2004), and the Wetness Index based on Landscape position and Topography (WILT) (Meles et al., 2020). Some topography-based indices use stream networks in the calculations, which are derived from flow accumulation grids such as the Depth to Water index (DTW) (Murphy et al., 2008) and Elevation Above Stream (EAS) (Rennó et al., 2008). Using this approach, streams are defined with a so-called stream initiation threshold, which is the accumulated area required to form

surface water. Selecting an appropriate stream initiation threshold has proven to be difficult due to temporal dynamics (Ågren et al., 2015) and soil textures (Ågren et al., 2014). Thresholds, such as stream initiation, used in terrain indices can be as, or even more, important as selecting the correct DEM resolution for the soil moisture modelling.

The use of airborne LiDAR has increased both the accuracy and resolution of DEMs and, as a result, soil moisture modelling (Murphy et al., 2011; Ågren et al., 2014; Leach et al., 2017; Kopecký et al., 2021). However, the resolutions of

DEMs used for hydrological modelling must reflect topographic features that are key elements in the hydrological response (Quinn et al., 1995). This means that higher resolutions do not necessarily result in better predictions, as the microtopography does not always control hydrological flow paths. Hence, there is a concern that the development of LiDAR-derived high-resolution DEMs has changed resolutions from being too coarse for small-scale hydrological modelling to being too high for many applications. With the use of terrain indices, there is often an optimal resolution depending on landscape type and specific

feature of interest (Gillin et al., 2015). Despite rapid LiDAR development, finding the optimal DEM resolution of terrain indices has remained relatively unexplored, with only a few exceptions (Seibert et al., 2007; Lin et al., 2010; Ågren et al., 2014).

In addition to DEM resolutions and user-defined thresholds, soil moisture modelling using terrain indices must take the local variations in soils and landforms into consideration. Across former glaciated landscapes, soil hydraulic properties are

often relatively consistent with unconsolidated ablation till overlaying basal till and/or bedrock. This means that hydrological pathways are significantly affected by the topography, resulting in soil moisture conditions in neighbouring areas differing greatly within short distances because of the local topography (Rodhe, 1987). The topographical effect on hydrological pathways is less pronounced in flat sorted sediment areas due to often low topographic variation and soils with consistent





hydrological conductivity at depth (Bachmair and Weiler, 2011). In landscapes with varying quaternary deposits, accurate soil
moisture predictions become more challenging (Güntner et al., 2004; Grabs et al., 2009; Zhu and Lin, 2011; Ågren et al., 2014), with consideration of these factors becoming important when interpreting modelled soil moisture.

      Recent promising approaches for accounting for landscape and soil variations have seen the combination of multiple terrain indices and other mapped information. One example of such an effort is the Swedish Soil Moisture Index (SMI) that combines DTW and the Soil Topographic Wetness index (STI) (Buchanan et al., 2013), and accounts for soil transmissivity
estimated from the quaternary deposit maps. An alternative is to use machine learning (Abowarda et al., 2021). Ågren et al. (2021) adjusted the soil moisture maps to local conditions over the whole of Sweden by training the model on field data from 16,000 plots and information from 28 maps. Key to this work were high resolution terrain indices calculated for different resolutions and thresholds. However, while machine learning is an excellent way of generating predictive models, it is difficult to interpret how the model combines indices with multiple resolutions and thresholds for different landscape types.

75       The aim of this study was, therefore, to evaluate how DEM resolution, thresholds and landscape types affect soil moisture predictions from a range of readily available terrain indices. We did this by examining which digital terrain index provided the best prediction of soil moisture classes within a heterogeneous but well-studied landscape in the boreal region, the Krycklan Catchment study (Laudon et al., 2020). Using a detailed forest and soil survey that covered the entire catchment allowed a test and performance evaluation of different terrain indices, in order to find the optimal resolutions and thresholds
for modelling soil moisture.

## 2 Methods

### 2.1 Site description

The 68 km$^2$ Krycklan Study Catchment is situated in the northern part of Sweden (Lat. 64°, 23´N, Long. 19°, 78´E) (Fig. 1). The catchment has a gentle topography, with a poorly weathered gneissic bedrock and elevations ranging from 127 to 372
m.a.s.l. The highest postglacial relict coastline crosses the area at around 257 m.a.s.l. The upper parts are dominated by glacial till while the lower parts are dominated by sorted sediments of sand and silt. Land cover is dominated by forest (87%) and a mosaic of mires (9%) and lakes. Forests are dominated by Scots pine (*Pinus sylvestris* L.) and Norway spruce (*Picea Abies* (L.) H. Kartst.) covering 63 and 26% respectively. Understory vegetation is dominated by ericaceous shrubs, consisting mostly of bilberry (*Vaccinium myrtillus*) and cowberry (*Vacinium vitis-ideae*) covering moss mats of *Hylocomium splendens* and
*Pleurozium schreberi*. Peatlands and wet areas have a vegetation dominated by *Sphagnum* species (Laudon et al., 2013). Forest soils are dominated by well-developed iron podzol. In addition to analysis over the entire catchment area, Krycklan was divided into two sub areas (till and sorted sediment) according to the quaternary deposits map, in order to analyse the effects of landscape types (Fig. 1).





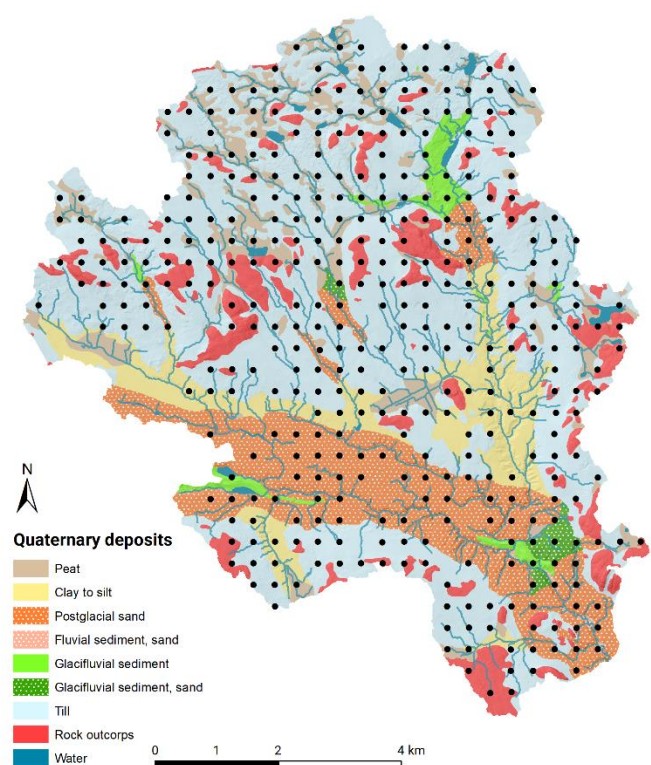

**Figure 1: The Krycklan catchment showing the quaternary deposits survey and plots as black circles.**

## 2.2 Forest survey

A forest survey grid was established in 2014, consisting of 500, 10 m radius survey plots (314.15 m$^2$) covering the entire Krycklan catchment, with each plot spaced 350 m apart. The survey plot locations were calculated using a randomly chosen origin and oriented along the coordinate axis of the Sweref 99 TM projection. Each nominal plot location was located in the field using a Garmin GPS 62stc GNSS receiver, and plot centres were marked with an aluminium profile. During a revisit, high accuracy centre positions were placed in the field using a Trimble GeoXTR DGPS receiver. Plots without high precision GPS locations, plots located on or outside the catchment boundaries, arable land, lakes and roads were excluded in this study. In total, soil moisture classifications were made for 398 plots during the autumn of 2014 and the spring of 2015.

## 2.3 Soil moisture field classification

Soil moisture conditions were determined for each survey plot in the field based on vegetation patterns and the position in the landscape, according to the Swedish national forest inventory (NFI) (Fridman et al., 2014). The soil moisture classes were: dry, mesic, mesic-moist, moist and wet.





- Dry soils have an average groundwater table more than 2 metres below the surface. Dry areas tend to be found on the
top of hills, ridges and eskers. The texture is often coarse and on flat terrain.

- Mesic soils are found on flat areas and on hillslopes. The average groundwater table is at a depth of 1–2 metres. It is
possible to walk across and keep dry feet, even directly after rain events or shortly after snowmelt. Soils are most
often podzols with a thin mor layer (4–10 cm) covered by mainly dry land mosses (e.g. *Hylocomium splendens*,
*Pleurozium schreberi*).

- Mesic-moist soils have an average groundwater table depth less than 1 metre. Mesic-moist areas are often located on
flat ground in lower lying grounds, or on lower parts of hillslopes. The soils tend to wet up on a seasonal basis.
Whether you can cross in shoes and remain with dry feet depends on the season and the time since the last heavy rain
or snow melting event. Patches of wetland mosses (e.g. *Sphagnum sp.*, *Polytrichum commune*) are common and trees
commonly tend to grow on humps. Podsol is commonly found, but with a thicker organic layer compared to mesic
sites. The organic layer is often classified as Peaty mor.

- Moist soils have an average groundwater table depth less than 1 metre. Surface water can be found in depressions.
Areas classified as moist are found at lower grounds, at the lowest parts of slopes and flat areas below larger ranges.
One can cross in shoes and keep one's feet dry by utilizing tussocks and higher lying areas. When stepping in
depressions, water will likely form around feet even after dry spells. The vegetation includes wetland mosses (e.g.
*Sphagnum sp.*, *Polytrichum commune*, *Polytrichastrum formosum*). Trees most often grow on small mounds and the
soil type is most often Histosol, Regosol or Gleysol.


- Wet soils are areas with permanent pools of surface water. These areas are often located on open peatlands. Drainage
conditions are very bad and it is not possible to cross these areas in shoes without ending up with wet feet. Where
trees occasionally occur, only in rare cases do they develop into stands. Soil type is most often Histosol or Gleysol.

**2.4 Digital terrain indices**

The study utilized a LiDAR-based Digital Elevation Model (DEM) created from an airborne laser scanning in August 2015.
A 0.5 x 0.5 m DEM was generated from a point cloud with 10 points per $m^2$. Horizontal and vertical errors were 0.1 m and 0.3
m, respectively. The DEM was resampled from 0.5 m to resolutions of 1, 2, 4, 8, 16, 32 and 64 m. Nine commonly used digital
terrain indices were calculated using DEMs with eight resolutions of 0.5, 1, 2, 4, 8, 16, 32 and 64 m (Table 1). The indices
Depth to Water (DTW) and Elevation Above Stream (EAS) use extracted stream networks in their calculations, with the size
of the stream network being set by the stream initiation threshold. For each resolution, DTW and EAS were calculated for the
stream initiation thresholds 1, 2, 4, 8, 16 and 32 ha, which is the range of the expected variability in the study region. The



Downslope Index (DI) was calculated with vertical distances of 2 and 4 m. A total of 146 terrain indices maps of soil moisture were produced. Field plot centre values for all indices maps were extracted for evaluation. All of the digital terrain indices were calculated using Whitebox Tools (Lindsay, 2016b), an open source program developed at the University of Guelph,
Canada. The code for aggregating the DEM and the Python code for the calculations can be found in the Supplementary material 1 (S1). In addition to the terrain indices that we calculated from the DEM, we also used two soil moisture maps downloaded from external sources: the SLU soil moisture map (Ågren et al., 2021) and a Soil Moisture Index map (SMI) developed by the Swedish Environmental Protection Agency (Naturvårdsverket, 2021).

**Table 1. All indices calculated in this study. Calculations were made for resolutions of 0.5, 1, 2, 4, 8, 16, 32, 64 m.[1]Calculations were also made for stream initiation thresholds of 1, 2, 4, 8, 16, 32 ha.[2] was calculated with vertical distances of 2 and 4 m. All GIS calculations were carried out using Whitebox Tools (Lindsay, 2016b), except for SLU soil moisture map and SMI which were downloaded from other sources.**

| Digital terrain indices | Abbreviation | Total number of layers |
|---|---|---|
| Topographic wetness index | TWI | 8 |
| Depth to water | DTW[1] | 48 |
| Elevation above stream | EAS[1] | 48 |
| Downslope index | DI[2] | 16 |
| Wetness index based on landscape position and topography | WILT | 8 |
| Relative topographic position | RTP | 8 |
| Plan curvature | PlanC | 8 |
| SLU soil moisture map | SLU | 1 |
| SMI | SMI | 1 |

**2.5 DEM pre-processing and extraction of stream networks**

Prior to hydrological modelling, the DEM was pre-processed to make it hydrologically accurate using the two-step breaching approach suggested by Lidberg et al. (2017). This approach works by first carving a short path into the DEM at locations where culverts and previously mapped streams intersect road embankments. Remaining depressions were resolved by a complete breaching approach using the tool Breach depressions in Whitebox Tools (Lindsay, 2016a). Two flow pointer grids and flow





accumulation grids (FA) were extracted from the hydrologically corrected DEM using the D-infinity flow routing algorithm (D∞) (Tarboton, 1997) and the multiple flow direction algorithm (MD∞) (Seibert and McGlynn, 2007). D8 (O'Callaghan and Mark, 1984) and D∞ (Tarboton, 1997) are both commonly used and widely implemented flow routing algorithms. MD∞ is an attempt to combine these two approaches and disperses flow like D∞ up to a user-defined threshold (aiming to simulate diffuse groundwater flows), after which it switches to operate like D8 without dispersion (aiming to simulate channelized flow of

surface waters). Stream networks were extracted from the flow accumulation raster using stream initiation thresholds 1, 2, 4, 8, 16 and 32 ha. Streams during different conditions can be mapped by varying the stream initiation thresholds. Larger stream initiation thresholds represent streams during low flow conditions while smaller thresholds represent conditions at high flow rates.

### 2.5.1 Topographic wetness index (TWI)

TWI predicts soil moisture based on local slope and the area's specific catchment area (Eq. 1), where $\alpha$ is the specific catchment area and $\beta$ is the slope of the grid cells in degrees (Beven and Kirkby, 1979).

$$T_{WI} = \ln(\alpha/tan\beta) \tag{1}$$

This was calculated using the D∞ flow algorithm for all eight DEM resolutions.

### 2.5.2 Depth to Water (DTW)

The Depth to Water index predicts soil moisture using the surface water source grid (stream network) and the surrounding landscape (Murphy et al., 2008). The DTW index refers to the least-cost path from any cell in the landscape to the nearest surface water cell (DTW = 0) channel. DTW is expressed as Eq. (2), where $dz_i$ and $dx_i$ represent the vertical distance between two cells.

$$D_{TW} = \left[ \sum \frac{dz_i}{dx_i} a \right] x_c \tag{2}$$

The constant $a$ is equal to 1 if the path between the cells connects parallel to the cell boundaries or $\sqrt{2}$ if it connects the cell diagonally; $x_c$ is the size of the raster cells. Cells located far away or at higher elevation from the flow channels will have high DTW values, meaning that the cells are drier. Stream cells were calculated using the source layers with extracted streams from the (MD∞) pointer described above. DTW was calculated for each of the six stream initiation thresholds and eight DEM resolutions.






### 2.5.3 Elevation Above Stream (EAS)

EAS indicates soil moisture by using the source layer with extracted streams described above and the original DEM (Rennó et al., 2008). EAS is calculated from the elevation difference between a grid cell in the landscape and the nearest stream cell calculated from the nearest flow path from the (MD∞) pointer grid. EAS was calculated for each of the six stream initiation
thresholds and eight DEM resolutions.

### 2.5.4 Downslope index

The downslope index represents the length of a flow path required to drop a given vertical distance $d$ (m) (Eq. 3 and Eq. 4) (Hjerdt et al., 2004). The algorithm calculates the distance downslope required to travel in order to descend $d$ metres. Downslope index can be reported both as a distance $d$ and a gradient, $\tan \alpha_d$, where the horizontal distance to the point $d$ metres
below follows the steepest directional flow path.

$$tan\, \alpha_d = \frac{d}{L_d} \tag{3}$$

Local linear interpolation is used between the two points to calculate the value of $L_d$. The slope angle between the starting point and the target point is represented by $\alpha_d$. For elevation differences approaching zero, the values of $\tan\alpha_d$ approach the local ground surface gradient, $\tan\beta$:

$$\lim_{d \to 0} tan\alpha_d = \tan\beta \tag{4}$$

Downslope index was calculated for 2 m and 4 m as the given vertical distances.

### 2.5.5 Wetness Index based on Landscape position and Topography (WILT)

WILT assumes that soil moisture is inversely proportional to $\Delta X$ and $\Delta Z$ in a groundwater-dominated landscape, where $\Delta Z$ is the depth to groundwater and $\Delta X$ is the horizontal distance to the nearest surface water feature (Eq. 5) (Meles et al., 2020).
WILT is a modification of TWI, obtained by dividing the upslope contribution area A by $\Delta X$ and $\Delta Z$:

$$WILT = \ln\left(\frac{A}{\Delta X * \Delta Z * \tan\beta}\right) \tag{5}$$

In this study, we calculated WILT where A was the upslope source area using D∞ flow accumulation, as with the TWI calculations. $\Delta X$ was derived from the downslope distance to stream and lakes using surface waters. $\Delta Z$ was the elevation difference between the DEM and modelled groundwater, represented by a DTW calculated for the property map.





### 2.5.6 Relative Topographic Position (RTP)

RTP is an index for the local position of a point in the landscape relative to its surroundings, which accounts for elevation distribution (Eq. 6). Within a user-specified local neighbourhood size, the RTP function uses the central elevation relative to the minimum ($z_{min}$), mean ($\mu$) and maximum elevation ($z_{max}$):

$$RTP = \frac{(z_0 - \mu)}{(\mu - z_{min})}, if \ z_0 < \mu \ \ or \ \ RTP = \frac{(z_0 - \mu)}{(\mu - z_{max})}, if \ z_0 >= \mu \tag{6}$$

RTP index is bound by the interval of [-1, 1], indicating whether the cell is above or below the filtered mean (Newman et al., 2018).

### 2.5.7 Plan Curvature (PlanC)

The plan curvature represents the curvature of the surface perpendicular to the direction of the slope direction (Wilson and Gallant, 2000). This index shows the divergence and convergence of slopes, where values are positive for convergent areas and negative for divergent ridges. The plan curvature was chosen for its influence on the downslope convergence and divergence of water flow paths.

### 2.5.8 Soil Moisture Index (SMI)

We also included an SMI from the national land cover database of the Swedish Environmental Protection Agency (Naturvårdsverket, 2021). This SMI was calculated as (Eq. 7):

$$SMI = \left(0.7 \times \frac{1}{DTW}\right) + (0.3 \times STWI) \tag{7}$$

This is a weighted map combining DTW and a modified TWI calculation, the Soil Topographic Wetness Index (STWI) (Buchanan et al., 2013), which accounts for soil transmissivity estimated from the quaternary deposit maps.

### 2.5.9 SLU Soil moisture map

A recent development in soil moisture mapping has been the use of machine learning to combine multiple soil moisture indices into one map (Lidberg et al., 2019; Abowarda et al., 2021; Ågren et al., 2021). Ågren et al. (2021) developed a new soil moisture map of Sweden by utilizing a variety of nationwide information, including the above-mentioned terrain indices, climate data and quaternary deposits. Training data consisted of nearly 16,000 field plots spread across the Swedish forested landscape from the national forest inventory. The final map showed the probability (0–100%) of a soil being wet. The SLU soil moisture map was produced at 2 m resolution.


### 2.6 Statistics

#### 2.6.1 Orthogonal Projections to Latent Structures (OPLS)

To ascertain which digital terrain index provides the best prediction of soil moisture within this heterogeneous landscape, we used Orthogonal Projections to Latent Structures (OPLS) analysis. Field classifications of soil moisture at each plot were used to evaluate the terrain indices through direct plot by plot comparison. OPLS was carried out on the entire catchment. The OPLS was carried out using the multivariate statistical program SIMCA 16.0, Umetrics, Umeå. The method of OPLS is a modification of Partial Least Squares regression (PLS) (Eriksson et al., 2006). In OPLS, the systematic variation in the predictors (X) is divided into two parts: one part that is predictive for the determinant (Y) (in this case, the field-determined soil moisture classes) and the orthogonal i.e. not related to Y. OPLS produces a model with improved interpretability compared to the ordinary PLS method. The method is used to identify important variables for predicting Y and singling out less important variables containing "noise". High positive or negative loadings on the predictive axis (pq[1]) indicate variables that are, respectively, positively or negatively correlated with Y, with increased correlation further away from origin. The orthogonal axis shows how much of the variation for each variable was not correlated with the determinant (Y). Before analysis, all variables were transformed to fit normality using a log transformation in SIMCA. SIMCA 16.0 also calculates the influence of each X variable in the model called Variable Importance on Projection (VIP). VIP components of an OPLS model are VIP predictive, VIP orthogonal as well as VIP total component. The VIP values are regularized such that if all X variables had the same importance for the model, they would all take the value one. VIP values larger than one for either VIP component indicate X variables that are important for that part of the model (Eriksson et al., 2006). Analysis was carried out in SIMCA 16.0 and figures were produced using R version 4.0.2 (R Core Team, 2020) and the package ggplot2 (Wickham, 2016).

#### 2.6.2 Confusion matrix

To evaluate the effects of landscape type i.e. sorted sediment and till soils within the catchment, we used the terrain index that preformed best in the OPLS analyses and its correspondence to the two wettest soil moisture classes (wet and moist). The overall conformance of the best terrain index with the combined wet and moist classes was assessed using confusion matrixes, accuracy (ACC) (Eq. 8) and Matthews Coefficient (MCC) (Eq. 9). The confusion matrix consists of true positives (TP), so accurately predicted wet plots, and false positives (FP) where dry plots were predicted wet, true negatives (TN), where the map correctly predicted dry plots and false negatives (FN) where the map predicted dry areas on wet plots. Accuracy (ACC) was assessed for each of the plots by:

$$Acc = \frac{TP+TN}{TP+TN+FP+FN} \tag{8}$$

The confusion matrix was further evaluated using Matthews Correlation Coefficient (MCC), for which a value of 1 indicates a perfect fit, 0 no better than random predictions and -1 a perfect negative correlation. MCC was calculated as (Eq. 9):



$$MCC = \frac{TP \times TN - FP \times FN}{\sqrt{(TP+FP)(TP+FN)(TN+FP)(TN+FN)}} \qquad (9)$$

For unbalanced datasets such as this, MCC is the best measure of model performance (Boughorbel et al., 2017).

## 3 Results

### 3.1 Field data

The field survey showed that the dominant soil moisture class was mesic, making up 60% of the survey plots in the catchment (Table 2). Mesic-moist was the second largest class with 15% of the plots; the moist and wet classes each made up 8%. The driest class (dry) made up 10% of the total plots in the catchment. Dividing the catchment into till and sorted sediment areas using the quaternary deposit map (Fig. 1), the proportion of classes became substantially different. Only 6% of the plots in the sorted sediment area were classified as moist or wet, compared to 20% in the till areas. A larger percentage (12%) of plots

were found in the driest class in the sorted sediment areas compared to the till areas (9%).

**Table 2. Percentage of observations in the five soil moisture classes for the entire Krycklan catchment and divided into till and sorted sediment areas**

Soil moisture classes

| Soil moisture class | Dry | Mesic | Mesic – moist | Moist | Wet | Plots (n) |
|---|---|---|---|---|---|---|
| Entire catchment | 10% | 60% | 15% | 8% | 8% | 398 |
| Till | 9% | 57% | 15% | 10% | 10% | 293 |
| Sorted sediment | 12% | 69% | 13% | 3% | 3% | 105 |

### 3.2 OPLS analysis

The OPLS analysis loading plot showed large variation in performance within and between terrain indices (Fig. 2). Figure 2 only shows the variable names from the best resolution for each digital terrain index and threshold based on the $VIP_{predictive}$ value shown in Figure 3, as the graph would be too cluttered if all 146 variable names were displayed. There is an interactive plot in the Supplementary material 2 (S2) where the name of each variable can be found. The general patterns of the effects of scale and threshold are indicated by the size and colour of the dots in the OPLS loading plot (Fig. 2). In order to help the reader

to visualize the effects of scales and resolution, the indices and thresholds have been grouped together by using coloured guides to connect terrain indices moving from high to low resolutions.





The OPLS analysis demonstrates that the DTW was a strong predictor of soil moisture classes (Fig. 3), but only if the optimal resolution and stream initiation threshold were used (Fig. 2). DTW loadings were located below zero on the predictive axis due to a negative relationship to soil moisture classes. Generally, the DTW variables were clustered together according to

thresholds, with decreasing performance for coarser DEM resolutions. The loading of EAS followed the pattern of DTW and both terrain indices had the highest predictive performance at stream initiation thresholds of 1 and 2 ha in DEM resolutions of 1–4 m. The highest resolutions of 0.5 m had a lower predictive performance (Fig. 2). Increased stream initiation thresholds above 2 ha lowered the predictive performance and added noise, as shown on the orthogonal axis (poso[1]).

The SLU soil moisture and SMI maps both performed well and were the second and fourth best terrain indices,

respectively, for predicting soil moisture classes (Fig. 3). SMI scored lower on the predictive axis (pq[1]) and had slightly higher variation not related to the soil moisture classes compared to the SLU soil moisture map (Fig. 2). The SLU soil moisture map and DTW were the best performing soil moisture predictors and had a very similar VIP$_{predictive}$ value (Fig. 3).

Downslope index (DI) was shown to be a good soil moisture class predictor. DI was positively correlated to soil moisture classes and therefore located on the positive pq[1] axis. DI2m (d=2 m) performed better than DI4m (d=4 m) with

higher loading on the predictive axis and lower loading on the orthogonal axis. For both DI2m and DI4m, a resolution of 2 m had the highest predictive performance (pq[1]) with the lowest noise. Resolutions below 2 m and above 8 m reduced the performance of the predictions substantially.

The performance of TWI was highly sensitive to the resolution of the DEM, too fine or too coarse resolutions gave nonsensical results. For this landscape, 16 m was found to be the optimal resolution for TWI calculations (Fig. 3).

WILT showed the highest value on the positive predictive axis at 8 and 4 m resolution, which was also true for RTP. The WILT loadings were slightly higher than the best performing TWI on the predictive axis (pq[1]), but also much higher on the orthogonal axis, indicating a large variation not related to soil moisture (Fig. 2). RTP showed no clear clustering, similar to TWI, and performed worse compared to the above-mentioned terrain indices (Fig. 3). Plan curvature scored low on the horizontal axis indicating that this variable was not a good soil moisture predictor for this landscape, something also confirmed

by the VIP$_{predictive}$ value being below one.

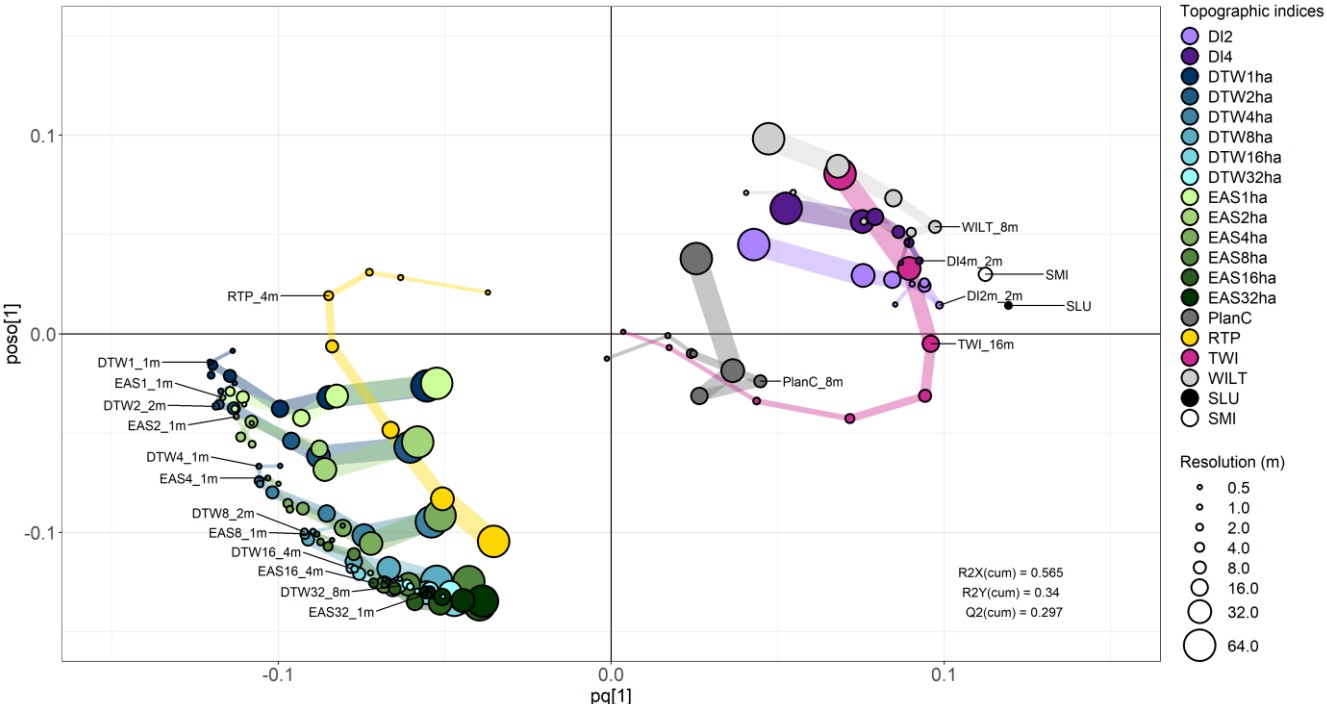

**Figure 2. OPLS loading plots for the Krycklan catchment and DEM-derived terrain indices in respect of soil moisture predictions. Variables that cluster closely within the same neighbourhood along the far sides of the horizontal axis are the more robust soil moisture predictors across DEM scales. Coloured guides connect terrain indices moving from small to large resolutions as depicted by the symbol size. In the loading plot, predictive performance increases with increased distance from 0 on the predictive axis (pq[1]). Negative and positive values on the (pq[1]) axis correspond to negative and positive correlations with Y. The orthogonal axis (poso[1]) represents how much of the variation for each variable was not correlated with the determinant (Y). For the reader who is interested in the details, we have published an electronic version of this graph where all labels are visible by moving the cursor on top of each circle (S2).**




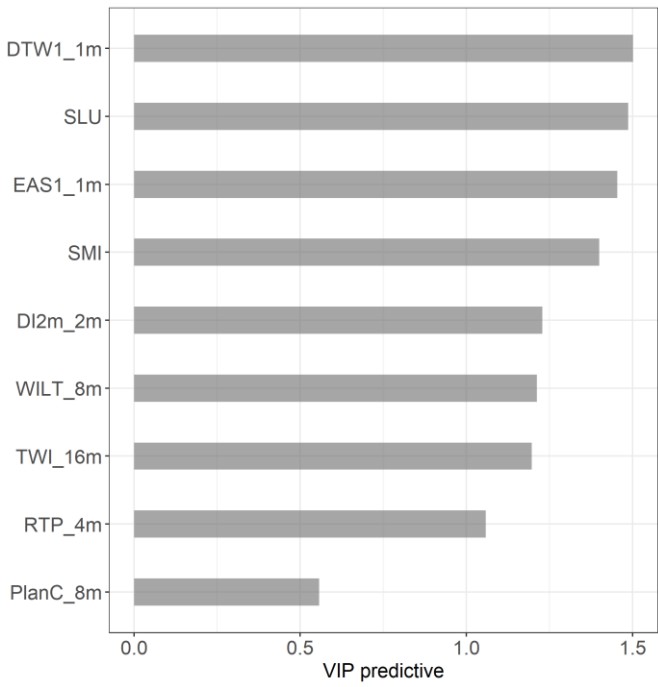


**Figure 3. VIP_predictive values for the best performing variable for each terrain index. In OPLS, VIP_predictive < 1 are variables that are better at explaining Y.**

### 3.3 Visual evaluation

Wet and moist soil conditions within the catchment are mostly found on mires, or as riparian soils along streams as shown in Figure 4. In the IR orthophoto (Fig. 4a), mires can be seen in the flatter areas (Fig. 4b) in the northern parts of the selected area. Several small stream channels in the bottoms of valleys drain the area from northwest to southeast, which borders onto wet riparian soils. Modelled soil moisture conditions of the best performing indices showed similar but varied agreement with natural features when visualized (Fig. 4), with DTW and SLU soil moisture maps clearly delineating the mire in the northwest

corner and around the lake, as well as the dryer hilltops in the southeast corner (Fig. 4). With the appropriate resolution and thresholds, many of the terrain indices were able to represent the variation of soil moisture conditions in more or less accurate ways after visual comparison. RTP had a poor performance in the OPLS, which is in line with the results demonstrated in Fig. 4j, where it predicted dry areas within the mire.



**Figure 4. Orthophoto (©Lantmäteriet) (a) and hill-shaded DEM (b) covering a till area within the Krycklan catchment. Below, maps of the highest performing maps of different terrain indices in order of VIP$_{predictive}$ with simplified common symbology for terrain indices (c–j) based on value distribution for the visual comparison.**

Figure 5 illustrates differences in the effects of increased DEM resolution represented by modelled results for TWI and DTW with a 1 ha stream flow initiation threshold. Varying DEM resolution had larger effects on the spatial variation of soil moisture using TWI compared to DTW, which was less affected; this is illustrated by the large differences moving from TWI at 1 m resolution to 32 m. The distribution of wet areas was not affected by DEM resolution for DTW compared to TWI. To visualize the effects of different user-defined stream initiation thresholds, DTW maps calculated for 1, 2, 4, 8, 16 and 32 ha stream



Hydrology and
Earth System
Sciences

Discussions

networks were created (Fig. 6). Increasing the stream flow initiation threshold shortens the stream network, resulting in a dryer

landscape model, and decreasing the stream flow initiation threshold models a wetter landscape.

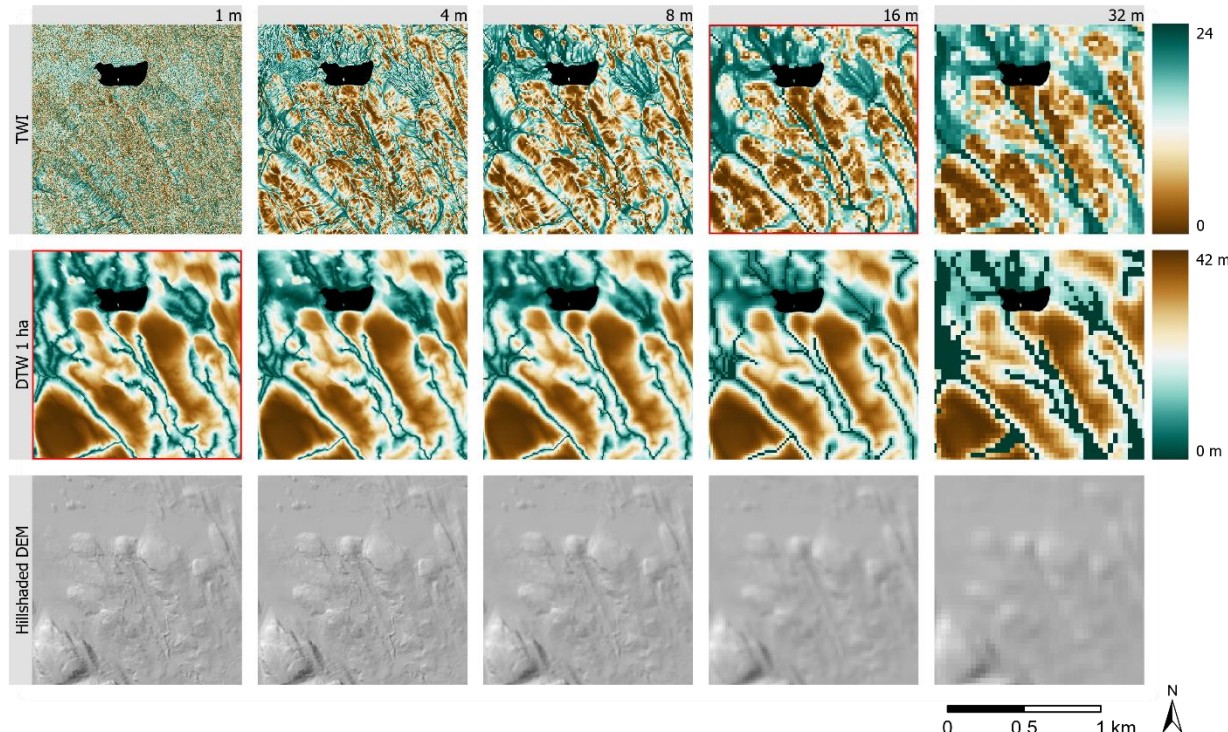

**Figure 5. Maps of TWI, DTW and hill-shaded DEM at 1, 4, 8, 16 and 32 m resolution. Best performing rasters in the OPLS analysis are outlined in red.**

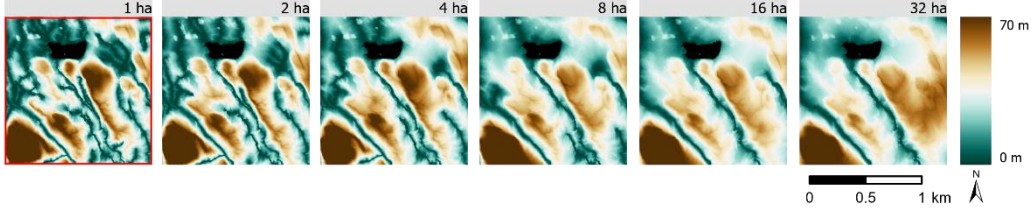

**Figure 6. Maps of DTW at 1 m resolution with stream initiation thresholds from 1 to 32 ha.**



## 3.4 Confusion matrix

The overall agreement of DTW 1m 1ha values (DTW<1) in relation to wet and moist soil classes was further tested using a confusion matrix (Table 3). Over the entire catchment area, the accuracy was 77% with a MCC of 0.42. Dividing the catchment into till and sorted sediment areas revealed significant differences in conformance. On the till area of the catchment, the DTW accuracy was higher with an accuracy of 78% and a MCC of 0.50. On the sorted sediment area of the catchment, DTW falsely predicted a large proportion of the dry plots as wet (FP). Only a third of the wet plots were predicted as wet (TP).

**Table 3. Confusion matrix of true positives (TP), true negatives (TN), false positives (FP) and false negatives (FN), representing Wet (Positive) and Dry (Negative) plots predicted by DTW 1 ha at 1 m resolution, as well as prediction accuracy (ACC (%)) and Matthews Correlation Coefficient (MCC (%)). Confusion matrixes and statistics were calculated for the entire catchment and divided into till and sorted sediment areas.**

| | Area | Plots (n) | TP | TN | FP | FN | ACC (%) | MCC |
|---|---|---|---|---|---|---|---|---|
| DTW | Entire catchment | 398 | 61 | 245 | 61 | 31 | 77 | 0.42 |
| | Till | 293 | 53 | 176 | 49 | 15 | 78 | 0.50 |
| | Sorted sediment | 105 | 8 | 69 | 12 | 16 | 73 | 0.20 |
| SLU | Entire catchment | 398 | 78 | 255 | 21 | 44 | 84 | 0.60 |
| | Till | 293 | 69 | 180 | 11 | 33 | 85 | 0.66 |
| | Sorted sediment | 105 | 9 | 75 | 10 | 11 | 80 | 0.34 |

## 4 Discussion

Modelling the spatial patterns of soil moisture remains an important scientific challenge and terrain indices are potentially a useful tool. As the availability and resolution of DEMs have increased, so have the uses of terrain indices for modelling hydrological, environmental and soil properties. However, the predictive performance of terrain indices is highly dependent on identifying the optimal spatial scales and user-defined thresholds for modelling soil moisture (Sørensen and Seibert, 2007; Lin et al., 2010; Ågren et al., 2014). The aim of this study, was therefore, to evaluate how DEM resolution, user-defined

thresholds and landscape types affect soil moisture predictions from a range of readily available terrain indices in relation to field-classified soil moisture conditions across a boreal catchment. Our results demonstrate the potential of terrain indices for modelling soil moisture when the optimal DEM resolutions and user-defined thresholds are selected.





Several terrain indices were able to predict the spatial variation of soil moisture classes within our study area, however our results revealed a large variation in the predictive performance within, and between, terrain indices at different DEM

resolutions and index-specific thresholds (Fig. 3). The general agreement of appropriately scaled terrain indices with field classified soil moisture conditions (Fig. 2) and visualized maps (Fig. 4) supports the underlying assumption that topography acts as the main driver of spatially varying soil moisture conditions. This is in line with many previous studies relating terrain indices to soil moisture conditions (Lin et al., 2010; Seibert et al., 2007; Grabs et al., 2009; Murphy et al., 2011; Ågren et al., 2014) and groundwater levels (Rinderer et al., 2014).

Ground truthing is required to evaluate the performance of different terrain indices, to prevent inappropriate choices of resolution and user-defined thresholds resulting in non-representative predictions of soil moisture. As its ground truth, this study used a uniquely extensive and high precision field survey within a well-studied landscape. We used field-mapped soil moisture classes based on vegetation patterns as a proxy for average soil moisture conditions, thus reducing the uncertainty associated with the large temporal and small-scale spatial variability of soil moisture (Murphy et al., 2011; Oltean et al., 2016;

Beucher et al., 2019; Lidberg et al., 2019). The vegetation patterns that form the basis for the classifications stay constant over time. In contrast, more direct soil moisture measurements using values such as soil water content and Time Domain Reflectometry (TDR) are greatly affected by the specific weather conditions before, and at the moment of, measurement. On the other hand, by using soil moisture classes as the ground truth, we only evaluate the "average" soil moisture conditions for each site and thereby focus on the spatial variability of relative soil moisture conditions within the landscape. We do, however,

acknowledge that soil moisture varies greatly with season and depends on regional weather conditions causing stream networks and wet soils to expand and shrink during the year (Ågren et al., 2015).

Our results highlight that the optimum DEM resolution for soil moisture predictions differed depending on terrain index, and further demonstrated the large effects of DEM resolution within certain terrain indices (Fig. 5). In line with previous studies, TWI was greatly affected by DEM resolution and was shown to perform best with a coarser 16 m resolution while

performing poorly with high resolution DEMs. This agrees well with previous studies both within (Sørensen and Seibert, 2007; Ågren et al., 2014) and outside the study area (Lin et al., 2010; Murphy et al., 2011). While this has previously been shown in the literature, a concern with the rapidly increasing numbers of high resolution DEMs worldwide is that researchers will use the most commonly used terrain index TWI, and disregard its poor performance with high resolution DEMs. Other indices, such as DTW, EAS and DI, had the best performance for resolutions between 1 and 4 m, and were stable within this range, as

shown for DTW in Figure 5. When using high resolution DEMs, the importance of selecting the optimal method for the pre-processing step (the hydrological corrections of depressions) increases. The impoundments caused by road banks (that are captured in high-resolution DEMs) otherwise cause problems for the subsequent steps of modelling the flow paths in the landscape (Woodrow et al., 2016; Leach et al., 2017). Studies have shown that it is better to pre-process the DEM using breaching, rather than filling functions (Wang et al., 2019; Lidberg et al., 2017). In this study, we used the protocol suggested

by Lidberg et al. (2017), as that study was carried out on similar glaciated catchments in Sweden. The highest resolution of DEM did not perform optimally for any of the evaluated terrain indices. This has been highlighted by previous studies to be





caused by small-scale variations in surface topography that do not affect the overall hydrological pathways (Gillin et al., 2015). The dependency of DEM resolution is important for any type of digital soil mapping and the optimum resolutions have been shown to be different, depending on landscape and the spatial scale of the environmental phenomena and processes involved
in the soil property of interest (Cavazzi et al., 2013).

This study also demonstrated the effects of adjusting user-defined thresholds associated with certain indices calculations (Fig. 2 & Fig. 6). In line with a previous study modelling the spatial extent of wet areas, DI calculated with a 2 m given vertical distance (d) (Eq. 3) performed best (Hjerdt et al., 2004). DTW and EAS were among the best performing terrain indices (Fig. 3), however the overall predictive performance was dependent on the chosen stream initiation thresholds (Fig. 2).
The best performance was achieved at 1 ha followed by 2 ha stream flow initiation threshold, much in line with previous results from the studied catchment area (Ågren et al., 2014), and from other study areas (Oltean et al., 2016). However, that only means that those thresholds might work well for glaciated catchments: in other regions, these thresholds might need to be adjusted. Our study highlights the need for ground truthing of the digital terrain indices, as the quality of the generated maps are so dependent on the selected thresholds. The substantial effects of varied user-defined thresholds for DTW and EAS
highlights the importance of caution when selecting terrain indices.

The unique setting of the Krycklan catchment, with its heterogeneous soils, made it possible for this study to demonstrate the challenges raised from variable landform types, where the assumption of topography acting as a first order control of soil moisture becomes less valid. In the sorted sediment areas of the Krycklan catchment, topographic variation is low and hydraulic conductivity high, allowing for deeper infiltration of water which decreases the topographical control of
groundwater flows compared to the upper till which dominates parts of the catchment (Jutebring Sterte et al., 2021). The layout of the study did not allow separate analysis of the different land form classes due to the limited number of field plots and low variation of soil moisture classes in the sorted sediment area (Table 1). However, by using a confusion matrix of the classified best performing terrain index (DTW < 1 m) from the OPLS in conformance with wet and dry soils, this study demonstrated a large difference in the MCC values between the sorted sediment (0.20) and till (0.50) parts of the catchment. The attempts of
combining terrain indices and other mapped information to tackle the challenges of soil moisture modelling faced by landscape heterogeneity did not outperform the more basic terrain indices **at** the entire catchment level. SMI and the SLU soil moisture map performed in a similar way to the DTW index, which is also the main component in their calculations. The confusion matrix using DTW's overall conformance clearly showed the challenges caused by the sorted sediment areas of the catchment (Table 3). This study highlights the necessity of adapting soil moisture predictions **to** local soil conditions when choosing
resolutions, terrain indices and their associated thresholds.

The results demonstrate the potential of terrain indices for modelling soil moisture across the landscape when the optimal scales and thresholds are selected for the calculations. Terrain indices have been related to soil properties (Seibert et al., 2007; Zajícová and Chuman, 2021), ecological studies (Zinko et al., 2005; Bartels et al., 2018) and site productivity (Mohamedou et al., 2017; Bjelanovic et al., 2018), and will likely develop further as a useful tool within many fields of study.
However, it should be recognized that the predictive power of terrain indices is limited by the non-topographical drivers of the

spatial variation in soil moisture which will always be significant, and rarely less than 50% (Western et al., 1999). Such drivers are, for example, soil depth, texture, hydrological conductivity, permeability and vegetation (Gwak and Kim, 2017). With an increasing demand for high-resolution spatial and temporal soil moisture models for climate, hydrology and soil modelling, it is important to understand underlying the covariate factors used to build them.

**5 Conclusion**

This study was designed to test, demonstrate and visualize the importance of appropriate scaling when modelling soil moisture using terrain indices. Although some previous studies have drawn similar conclusions, there is still a tendency within many fields to use the highest DEM resolution available when using terrain indices to represent soil moisture conditions as a covariate. However, one size - or resolution in this case - does not fit all. Due to the differences in climate, landscape types

and soil texture, terrain indices must be adapted to local conditions and calculated at appropriate scales and thresholds. Heterogeneous landscape types remain a challenge for predicting soil moisture conditions and should be taken into account when interpreting modelled results. We, therefore, stress the importance of evaluating the modelled terrain index results for the area of interest and not to extrapolate the optimum terrain indices for our study areas directly.

**6 Code and data availability**

The code and data used in this study for aggregating the DEM and generating the different terrain indices is available in Supplementary material 1 (S1).

**7 Supplement**

The supplementary material related to this article S1 and S2 is available online at:

https://data.mendeley.com/datasets/dg64p8wmj9/draft?a=3ca61bcf-627a-4a26-93ea-5b4327a8eaab

**8 Author contribution**

All authors were responsible for the conceptualization of the study and evaluation of results. WL was responsible for the data

curation and JL performed the analysis. JL prepared the manuscript and figures and lead the writing of the paper with contributions from all the co-authors.



## 9 Competing interests

The authors declare that they have no conflict of interest.



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
