# Peer review of "Predicting soil moisture conditions across a heterogeneous boreal catchment using terrain indices"

_Hydrology and Earth System Sciences, 2021_

## Author Response (AR1)

**Author's response to Referee 1**

**Referee #1 introductory comment:** The paper by Larson and colleagues deals with the important aspect of predicting soil moisture classes in a study catchment in Sweden. In general, the paper is well written and properly structured and the methods that the authors used to derive various terrain indices are sound. Also worth commending is the large number of field observations n = 398 which were used in the statistical analysis.

**Author's response**: We thank Referee #1 for the overall positive comments. Our responses to all the comments of Referee #1 are listed below in the order they appear.

**Referee #1:** A concern for me, however, is the use of the term 'soil moisture' throughout the manuscript. Soil moisture and soil moisture classes are not the same thing, in my opinion. Soil moisture is a soil property with both a spatial (lateral and vertical) and a temporal dimension. The soil moisture classes that the authors used (described in line 105 – 133) have only a spatial (and only lateral) dimension. It is therefore rather a mapping unit or a soil association than a soil property.

**Author's response:** We appreciate the concern raised by Referee #1 regarding the use of the term soil moisture. To clarify that we mean soil moisture conditions we suggest to change the title from "soil moisture" to "soil moisture conditions". We will also suggest clarifying the use of the term soil moisture/soil moisture conditions in the introduction and method section (see following comments).

Line 84-87: 'We did this by examining which digital terrain index provided the best prediction of field determined soil moisture classes within a heterogeneous but well-studied landscape in the boreal region, the Krycklan Catchment study (Laudon et al., 2020).'

**Referee #1:** The qualitative description of the soil classes also appears to be biased e.g., possible to walk over and keep dry feet…shortly after snowmelt. Surely this will depend on the size of the person and how long 'shortly' is. Also, the topographic descriptions of the soil moisture classes are raising questions about potential *circular reasoning*. How important is these topographic attributes in determining the soil moisture class? If they are a key determining factor, then surely you are not assessing whether the terrain indices are predicting soil moisture, but rather are the terrain indices able to predict *terrain indices*. So, in my opinion, the authors did not predict soil moisture, making the title and a lot of the discussion misleading.

**Author's response:** In relation to the previous comment Referee #1 raises concern about the definitions of the soil moisture classes. This is understandable in the current state of the manuscript. We have failed to describe that it is foremost the estimated average depth to groundwater during the vegetation period which determines the soil moisture class. This should be clearly stated in the first section of the description of the soil moisture field classification. To clarify this we suggest the following additions:

Line 117-122: 'Soil moisture classes were registered in the field following the protocol of the Swedish national forest inventory (NFI) (Fridman et al., 2014), based on an estimation of each plot's average depth to groundwater level during the vegetation period estimated from its

position in the landscape, vegetation patterns and soil type. This approach reduces the discrepancies caused by seasonal variation and provides an indicator of the general soil moisture conditions, which is the focus of this study. Survey plots were categorized into five classes dry, mesic, mesic-moist, moist and wet, which are described and presented below and can be found in more detail in the field instruction (Swedish NFI, 2014)'

Referee #1 also observantly raises concern regarding the potential of "*circular reasoning*". The depth to groundwater table was estimated with guidance from surrounding of topography, eventual presence of groundwater in small depressions, soil types and vegetation. This system is an established method and more detailed descriptions can be found in the field handbook of the Swedish national forest inventory which we now cite above. The descriptions in the text of the previous version are more of a general description of the soil moisture classes and not what determines the classification on site. We see our error here and hope that our adjusted version and added citation will be satisfactory. Thanks to Referee #1 we suggest to further develop the short soil moisture class descriptions with emphasis on the groundwater level as follows:

Line 124-150:

- Dry soils have an average groundwater table more than 2 metres below the soil surface. They tend to be coarse-textured and can be found on hills, eskers and ridges. The soils are mainly leptosols, arenosols, regosols or podzols (with thin organic and bleached horizons).

- Mesic soils have an average groundwater table between 1-2 metres below the soil surface. Podzol is the dominating soil type with a thin (4-10 cm) organic mor layer covered mainly by dryland mosses (e.g. Pleurozium schreberi, Hylocomium splendens, Dicranum scoparium). They can be walked on dry-footed even directly after rain or shortly after snowmelt.

- Mesic-moist soils have an average groundwater table depth less than 1 metre. Mesic-moist areas are often located on flat ground in lower lying areas, or lower parts of hillslopes. The soils tend to wet up on a seasonal basis. Whether you can cross in shoes and remain with dry feet depends on the season and the time since the last heavy rain or snow melting event. Patches of wetland mosses (e.g. Sphagnum sp., Polytrichum commune) are common and trees commonly tend to grow on humps. Podzols are commonly found, but often with a thicker organic layer compared to mesic sites. The organic layer is often classified as peaty mor.

- Moist soils have an average groundwater table depth less than 1 metre. The groundwater table is often visible in depressions. Areas classified as moist are found on lower grounds, at the lowest parts of slopes and flat areas below larger hills. One can cross in shoes and keep one's feet dry by utilizing tussocks and higher lying areas. When stepping in depressions, water should form around the feet even after dry spells. The vegetation includes wetland mosses (e.g. Sphagnum sp., Polytrichum commune, Polytrichastrum formosum). Trees often grow on small mounds and the soil type is most often histosol, regosol or gleysol.

- Wet soils are areas where the groundwater table is close to the soil surface and permanent pools of surface water are common. These areas are often located on open

peatlands. Drainage conditions are very poor and it is not possible to cross these areas in shoes without ending up with wet feet. Coniferous trees only seldom develop into stands. The soil type is most often histosol or gleysol.'

Line 404-407: 'We used field-mapped soil moisture classes based on estimated depth to groundwater from the soil surface guided by surrounding topography and vegetation patterns as a proxy for average soil moisture conditions, thus reducing the uncertainty associated with the large temporal and small-scale spatial variability of soil moisture'

**Referee #1:** With this being said, the prediction of soil moisture classes (soil associations) is still novel, and the paper makes a contribution to international literature. An important outcome is that a finer detailed DEM does not necessarily imply better predictions of soil moisture classes.

**Author's response:** We thank Referee #1 for these positive comments

**Referee #1:** The article would benefit by adding an improved description of the climate under section.

**Author's response:** We agree with Referee #1 that the paper would benefit from a more detailed description of the climate. We suggest to add the following three sentences:

Line 94-97: 'The climate is characterized as a cold temperate humid type with persistent snow cover during the winter season (Laudon et al., 2020). The 30 year mean annual temperature (1986-2015) is 2.1°C, with the highest monthly mean temperature in July and lowest in January (14.6 and -8.6 respectively). The mean annual precipitation equals 619 mm where more than 30% falls as snow.'

**Author's response to Referee 2**

**Introduction comment from Referee #2:** The manuscript rigorously compared the prediction of soil moisture maps via Digital terrain analysis. To this end, the authors made predictions using nine different terrain indices in combination with the available soil wetness maps, at varying resolutions and user-defined thresholds, with a field dataset of soil moisture registered in five classes from a forest survey covering a boreal landscape in Sweden.

It's a cliched but intriguing research topic with new findings that perfectly align with the journal's scope. The manuscript is generally well-written; however, brevity and flow are missing which makes it rather verbose. The artwork is legible and flawless. However, I have some minor concerns regarding some sections.

**Author's response:** Thank you Referee #2 for providing constructive criticism and suggestions which will improve the next version of the manuscript. In regards to the flow of the brevity of the text we will adjust this in the revised manuscript. Our responses to all the comments of Referee #2 are listed below. All comments from Referee # 2 have been included in this document, followed by our response.

**Referee #2:** The manuscript neither identifies the gap broadly in the scientific knowledge nor adds new knowledge to the overall body of scientific understanding. The novelty is based on the fact two recent studies (Abowarda et al., 2021; Ågren et al., 2021) of the same nature study area (Sweden) were flawed because the selected model for high-resolution terrain indices was restricted to 16000 plots and 28 maps, while the latter used machine learning which is prone to a vague combination of multiple resolutions and thresholds. Please further elaborate and align the flaws in the direction of gaps which you have covered by this study. Please further elaborate and align the flaws in the direction of gaps which you have covered by this study.

**Author's response:** In regards to the comments from Referee #2 about the identification of the knowledge gap addressed in this study we understand the referee's remarks. Referee #2 mentions two studies using Machine learning for soil moisture mapping, one across Sweden (Ågren et al. 2021) and one in China (Abowarda et al. 2021). Thanks to the remarks from Referee #2 we realize that we should clarify the knowledge gap further and the aim of our study. In regards to the above mentioned studies we see that in the growing field of machine learning there is a need of more process-based understanding of the underlying implications that DEM resolution, user-defined thresholds and parent material have on different terrain indices before making the selection of terrain indices to use. Choosing the wrong index, resolution or threshold can result in incorrect predictors, which we show in our study. With this study and approach using a small study area we can visualize the need to test the performance of terrain indices within any study area of interest. The usefulness and optimal resolutions and user-defined thresholds will depend on the landscape one chooses to study. Whilst the studies cited have accurate mapping of soil moisture as the aim, we wanted to understand the underlying effects of the variables often put into soil moisture mapping models. We suggest to add the following sentences in the revised manuscript:

Line 79-82: ' Due to the large applications, wide uses and availability of terrain indices there is a need of understanding the underlying effects of DEM resolution, user-defined thresholds and landscape types have on the modelled results. Using terrain indices to model soil moisture conditions on inappropriate scales and landscape types may result in inaccurate predictions.'

**Referee #2:** The abstract is general, please explain your robust findings to the readers as a take-home message.

**Author's response:** We have revised our abstract and suggest the following:

Line 7-23'**Abstract.** Soil moisture has important implications for drought and flooding forecasting, forest fire prediction and water supply management. However, mapping soil moisture has remained a scientific challenge due to forest canopy cover and small-scale variations in soil moisture conditions. When accurately scaled, terrain indices constitute a good candidate for modelling the spatial variation of soil moisture conditions in many landscapes. In this study, we evaluated seven different terrain indices at varying digital elevation model (DEM) resolutions and user-defined thresholds as well as two available soil moisture maps, using an extensive field dataset (398 plots) of soil moisture conditions registered in five classes from a survey covering a (68 km$^2$) boreal landscape. We found that the variation in soil moisture conditions could be explained by terrain indices and the best predictors within the studied landscape was Depth to water index (DTW) and a machine learning generated map. Furthermore this study showed a large difference between terrain indices in the effects of changing DEM resolution and user-defined thresholds which severely affected the performance of the predictions. For example the commonly used Topographic wetness index (TWI) performed best on a resolution of 16 m while TWI calculated on higher than 4 m DEM resolutions gave inaccurate results. In contrast the Depth to water (DTW) and Elevation above stream (EAS) were more stable and performed best on 1-2 m DEM resolution. None of the terrain indices performed best on the highest DEM resolution of 0.5 m. In addition this study highlights the challenges caused by heterogeneous soil types within the study area and shows the need of local knowledge when interpreting the modelled results. The results from this study clearly demonstrate that when using terrain indices to represent soil moisture conditions, modelled results need to be validated, as selecting unsuitable DEM resolution or user-defined threshold can give ambiguous and even incorrect results. '

**Referee #2:** Please explain the freezing temperature range, duration, minimum tree height and canopy cover as this feature explain the boreal forests.

**Author's response:** We agree with Referee #2 that the paper would benefit from a more detailed site description. We suggest to add the following two sentences in the site description:

Line 98-99: 'Due to forest management, Krycklan is a complex mosaic of forest stands of different age classes and species composition.'

Line 94-97: 'The climate is characterized as a cold temperate humid type with persistent snow cover during the winter season (Laudon et al., 2020). The 30 year mean annual temperature (1986-2015) is 2.1°C, with the highest monthly mean temperature in July and lowest in January (14.6 and -8.6 respectively). The mean annual precipitation equals 619 mm where more than 30% falls as snow.'

**Referee #2:** The discussion must be strong enough to support your findings. In its current form, it way weak and only convinces the readers regarding the consistency of findings with previously observed results. Please focus on the differences in climate, landscape types and soil texture, and terrain indices.

**Author's response:** Thank you Referee #2 for providing constructive criticism and suggestions which will improve the next version of the manuscript. We suggest adding the following sentences to clarify the implications of our findings in relation to previous studies.

Line 392-394: 'No previous study has been able to provide such detailed data at catchment scale, amount of terrain indices in combination with an extensive field survey which clearly demonstrates the importance of selection of terrain index, DEM resolution and index-specific thresholds.'

Line 419-424: 'However, this is in contrast to a recent study by (Riihimäki et al., 2021) where they thoroughly investigated the effect of DEM resolution and flow accumulation algorithms on TWI calculations in a 300 ha area of the northwestern Fennoscandian mountain tundra. Their conclusion was that Dinf reached its maximum explanatory power at 3 m resolution. This highlights that the optimal DEM resolution for predicting soil moisture conditions using TWI cannot just be taken from literature as it varies from site to site and it's necessary to investigate the optimal resolution for each landscape.'

Line 461-467: This study highlights the necessity of adapting soil moisture predictions to local soil conditions. These underlying factors need to be taken into consideration when modelling soil moisture conditions on any level from catchment, regional and national scale. One such attempt was the SLU soil moisture map which was constructed for the entire country of Sweden using vast amounts of field data from 16 000 field-plots across the country as training data and several digital terrain indices at multiple resolutions and thresholds. Even so, when evaluated on the Krycklan catchment the SLU soil moisture map ranked second among the top predictors for soil moisture (Figure 2 and 3) and did not outperform several of the more simple terrain indices.

Line 484: We, therefore, stress the importance of evaluating the modelled terrain index results for the area of interest and not extrapolating the optimum terrain indices for our study areas directly or to blindly use the DEM of the highest resolution available.

**Referee #2:** Figs. 2 and 3 are excellent but I would suggest combining these two figures for convenience. It will help the readers to understand the performance of each index. Similarly, please explain all details of the concerned figure once and all, the switching confuses the readers.

**Author's response:** We understand the remarks from Referee #2, however we believe figure 2 already contains the amount of information that a reader can grasp. The figures would also need to be named a) and b) and, would therefore in our opinion, not change the issue raised by the referee. We leave it up to the editor to decide if the figures are sufficient as they are in the preprint.